

# Comparative study of gut microbiota in Tibetan wild asses (*Equus kiang*) and domestic donkeys (*Equus asinus*) on the Qinghai-Tibet plateau

Hongjin Liu[1,2,3], Xinquan Zhao[1,2], Xueping Han[1,2,3], Shixiao Xu[1,2], Liang Zhao[1,2], Linyong Hu[1,2], Tianwei Xu[1,2], Na Zhao[1,2], Xiaoling Zhang[1,2,3], Dongdong Chen[1,2], Fuquan He[1,2] and Xin Chen[1,2,3]

[1] Northwest Institute of Plateau Biology, Chinese Academy of Sciences, Xining, Qinghai Province, China
[2] Key Laboratory of Adaptation and Evolution of Plateau Biota, Chinese Academy of Sciences, Xining, Qinghai Province, China
[3] University of Chinese Academy of Science, Beijing, China

Corresponding author
Shixiao Xu, sxxu@nwipb.cas.cn

## ABSTRACT

Tibetan wild asses (*Equus Kiang*) are the only wild species of perissodactyls on the Qinghai-Tibet Plateau and appears on the International Union for Conversation of Nature (IUCN) 2012 Red List of threatened species. Therefore, understanding the gut microbiota composition and function of wild asses can provide a theoretical for the situ conservation of wild animals in the future.In this study, we measured the dry matter digestion by the 4 molar hydrochloric acid (4N HCL) acid-insoluble ash method and analyzed the intestinal microbiota of wild asses and domestic donkeys by high-throughput sequencing of the 16s rDNA genes in V3–V4 regions. The results showed that the dry matter digestion in wild asses was significantly higher than in domestic donkeys ($P < 0.05$). No significant difference in alpha diversity was detected between these two groups. Beta diversity showed that the bacterial community structure of wild asses was acutely different from domestic donkeys. At the phylum level, the two dominant phyla *Bacteroidetes* and *Firmicutes* in wild asses were significantly higher than that in domestic donkeys. At the genus level, *Ruminococcaceae_NK4A214*, *Phascolarctobacterium*, *Coprostanoligenes_group*, *Lachnospiraceae_XPB1014_group* and *Akkermansia* in wild asses were significantly higher than in domestic donkeys. Moreover, statistical comparisons showed that 40 different metabolic pathways exhibited significant differences. Among them, 29 pathways had richer concentrations in wild asses than domestic donkeys, mainly included amino acid metabolism, carbohydrate metabolism, and energy metabolism. Of note, network analysis showed that wild asses harbored a relatively more complex bacterial network than domestic donkeys, possibly reflecting the specific niche adaption of gut bacterial communities through species interactions. The overall results indicated that wild asses have advantages over domestic donkeys in dry matter digestion, gut microbial community composition and function, and wild asses have their unique intestinal flora to adapt high altitudes on the Qinghai-Tibet plateau.

## INTRODUCTION

Low temperatures and hypoxia render the Qinghai-Tibet Plateau (QTP) an extremely harsh environment for the survival of mammalian species. The Tibetan wild asses (*Equus kiang*)—order Perissodactyla, family Equidae, genus *Equus*—is considered a unique species and the only wild species of perissodactyls to inhabit the QTP (*Zheng & Gao, 2000*), and appears on the International Union for Conversation of Nature (IUCN) 2012 Red List of threatened species. Extensive research has been conducted to promote the conservation of this species (*Dong et al., 2015*; *Guo et al., 2018*). Domestic donkeys (*Equus asinus*)—also of the order Perissodactyla, family Equidae, genus *Equus*—are the subspecies of wild ass (*Moehlman, 2002*) and were introduced from regions of China at lower elevations, adapting to life in this harsh environment and becoming the livestock of the nomadic Tibetan people to their substantial economic benefit. The intestines of Tibetan wild asses (TWAs) and domestic donkeys contain complex microorganisms that play an important role in converting indigestible plants into the energy needed for host growth, serving a bridge between forage digestion and host energy absorption. Therefore, the study of intestinal microorganisms is of great significance for revealing host immunity, nutrient metabolism, energy absorption and wild animal conservation.

Numerous scientific reports suggest that gut microbiota constitute a complex ecosystem involving a symbiotic relationship with the host that plays an important role in the host's performance, health and disease, nutrient absorption, and metabolic status (*Hintz & Cymbaluk, 1994*; *O'Hara & Shanahan, 2006*; *Sekirov et al., 2010*; *Wardwell, Huttenhower & Garrett, 2011*). To date, this research has focused mainly on humans, ruminants, marine organisms, and some monogastric animals to investigate the factors that affect intestinal microbiota, gut microbes and health, and gut microbes and nutritional metabolism (*Bian et al., 2016*; *Sekirov et al., 2010*; *Shepherd et al., 2012*). As herbivores, donkeys utilize forage nutrition by microorganism fermentation in the gastrointestinal tract. *Zhao et al. (2016)* studied bacterial diversity in the ceca of Xinjiang donkeys and found that *Ruminococcaceae* and *Lachnospiraceae* played a key role in digesting roughage feed. Due to the difficulty of obtaining samples from wild protected animals, research on gut microbes in TWAs is limited. A recent study compared the gut microbiota composition of wild and captive TWAs, revealing that captivity reduced intestinal microbial diversity and increased the risk of epidemics, thereby negatively impacting the health of wildlife (*Gao et al., 2019*). However, few studies have compared the intestinal microbial composition and function of TWAs and natural pasture domestic donkeys (NPDDs) on the QTP.

As members of the same genus—*Equus*—TWAs and NPDDs provide ideal candidates for the study of gut microbiota. Therefore, we used high-throughput sequencing of the V3-V4 regions of the 16s rRNA gene to investigate the composition and predict the metabolic pathways and functions of the gut microbes in TWAs and NPDDs. We hypothesized that TWAs were superior to NPDDs in terms of the composition and function of their intestinal flora. We think our study is of importance for the study of intestinal flora adaptability of Tibetan wild asses at the high altitude.

## MATERIAL AND METHODS

In this study, all animal care procedures were consistent with the guidelines from the Institution of Animal Care and Ethics Committee of the Northwest Institute of Plateau Biology, Chinese Academy of Sciences (NWIPB20160302), besides this project was carried out with the permission of local government (Qinghai provincial sciences & technology department). Chaoyuan Zhang the deputy director of the China Qinghai provincial science & technology department and Xinquan Zhao the academic dean of Northwest Institue of Plateau Biology Chinese Academy of Sciences gave these verbal authorizations to allow us to carry out the experiment here. So official documents and field permits are not required.

### Sample collection

A total of 30 fecal samples were obtained from two different sites. The TWAs group included 15 fecal samples from a swamp meadow (101°.06′E, 38°.37′N) in Qumarleb County (Yushu Prefecture, Qinghai Province, China) at an altitude of 4,300 m in August 2017; the NPDD group included 15 fecal samples from an alpine meadow (94°48′E, 34°.99′N) in Menyuan County (Haibei Prefecture, Qinghai Province, China) at an altitude of 3,200 m in August 2017. All of the laboratory animals were healthy and none of the donkeys had been administered antibiotics within the past three months. The fecal samples (approximately 2 ml each) were collected and transferred into separate sterilized tubes and stored immediately in liquid nitrogen for DNA extraction. The forage samples were collected from quadrats (50 cm × 50 cm) of grass on which the animals grazed. Twenty quadrats greater than 10 m apart were placed randomly to investigate the dominant species and collect the ground herbage. A total of 20 forage samples were collected from these two regions (10 per region). Forage samples were dried in a 65 °C oven for 24 h, then ground through a 1-mm sieve and stored in a vacuum dryer for nutritional analysis.

### Determination of plant nutrient composition

Total N was measured using the Kjeldahl method; the crude protein content was calculated as 6.25 × N (Method No. 984.13); the ether extract (EE) was measured using the Soxhlet system (Method No. 954.02); the acid detergent fiber (ADF) and neutral detergent fibre (NDF) were analyzed using the method described by *Soest, Robertson & Lewis (1991)*; the non-fibrous carbohydrate (NSC) was calculated as follows: $NSC(\%) = 100 - Ash(\%) - EE(\%) - NDF(\%) - CP(\%)$. The nutrition of the herbage and dominant species are shown in Table 1.

### Acid-insoluble ash to determine dry matter digestibility

The dry matter digestibility was measured by using a modified method of 4N-HCL acid-insoluble ash (AIA) (*Van Keulen & Young, 1997*). Briefly, 5 gram crushed forage and 5 gram dried excreta are boiled in 50ml 4N for thirty minutes; the slurry is then filtered by quantitative filtered paper (NEWSTAR®, Hangzhou, China, Φ 12.5 cm) and the residue washed free of acid using hot distilled water untilled the elute is neutral; the residue and the filter paper were ashed directly in crucible at 600 °C for 12 h in electric muffle furnace (Beijing zhongxing weiye instrument co. LTD, KSW-6-12); After

**Table 1  Comparison of the nutrient contents in the herbage composition of the TWAs and NPDDs groups.**

| Nutrient content of Herbage (%) | Groups[a] | | P value |
|---|---|---|---|
| | TWAs | NPDDs | |
| EE[b] | $1.40 \pm 0.10$ | $1.32 \pm 0.02$ | 0.09 |
| CP | $11.13 \pm 1.02$ | $10.96 \pm 0.43$ | 0.09 |
| ADF | $22.51 \pm 1.02$ | $28.50 \pm 1.30$ | 0.76 |
| NDF | $44.56 \pm 3.06$ | $48.82 \pm 1.87$ | 0.44 |
| NFC | $27.80 \pm 3.00$ | $25.39 \pm 2.02$ | 0.54 |
| Ash | $15.12 \pm 1.45^{a}$ | $13.54 \pm 0.34^{b}$ | 0.03 |
| Dominant species of herbages | *Kobresia tibetica, Kobresia pygmaea* | *Kobresia humilis, Kobresia capillifolia* | |

**Notes.**

[a]TWAs refers to Tibetan wild asses, NPDDs refers to natural pasture domestic donkeys.

[b]EE, ether extract; CP, crude protein; ADF, acid detergent fiber; NDF, neutral detergent fiber; NFC, Non-fibrous carbohydrate. Values in the same row with different superscripts are significantly different ($P < 0.05$).

the residue and filter paper complete combustion, the crucible and ash were placed in a dryer to cool to room temperature and then weighed on an analytical balance. The equation used to calculate dry matter digestibility was as follows (*Kavanagh et al., 2001*):

Dry matter digestibility $= 100 - 100 \times$ AIA in herbage / AIA in faeces.

## DNA extraction and purification

Microbial DNA was extracted from stool samples using the E.Z.N.A® Stool DNA Kit (Omega Bio-Tek, Norcross, GA, USA) according to the manufacturer's protocols. DNA quality was assessed via 2% agarose gel electrophoresis and metagenomic DNA concentrations were determined with a NanoDrop™ 2000 (Thermo Fisher Scientific, Waltham, MA, USA). The 16S rDNA V3-V4 regions of the ribosomal RNA gene were then amplified by PCR (95 °C for 2 min, followed by 27 cycles at 98 °C for 10 s, 62 °C for 30 s, and 68 °C for 30 s, with a final extension at 68 °C for 10 min) using the following primers: 341F, CCTACGGGNGGCWGCAG; 806R, GGACTACHVGGGTATCTAAT. The 5′-end of 314F primer includes 8-bp unique barcodes, which were used to split each sample. PCR reactions were performed in triplicate with a 50 μl mixture containing 5 μl of 10 × KOD buffer, 5 μl of 2.5 mM dNTPs, 1.5 μl of each primer (5 μm), 1 μl of KOD polymerase, and 100 ng of template DNA. The PCR products were detected by electrophoresis on a 2% agarose gel and purified using the AxyPrep DNA Gel Extraction Kit (Axygen Biosciences, Union City, CA, USA) according to the manufacturer's instructions and quantified using QuantiFluor®-ST (Promega, Madison, WI, USA). The purified amplicons were then pooled equimolarly for paired-end sequencing on an Illumina (San Diego, CA, USA) HiSeq 2500 platform (Guangzhou Gene Denovo Biotechnology Co Ltd Guangzhou, China) using standard protocols.

## Sequencing and processing

To get clean paired-end reads, the raw reads were filtered to remove reads containing greater than 10% unknown nucleotides (Ns) and fewer than 80% of base calls with quality scores less than 20. FLASH version 1.2.11 was used to merge paired-end reads as raw tags with

a minimum overlap of 10 bp and mismatch error rates of 2% (*Magoc & Salzberg, 2011*). The noisy raw tag sequences were filtered by QIIME (version 1.9.1) to obtain high-quality, clean tags (*Caporaso et al., 2010*). The filter commands mainly including tags interception and length filtering. That is, the raw tags were intercepted from the continuous low-quality value (the default quality threshold is <=3) to the first low-quality base site of the set length (the default length is 3); filtering out tags data that had a continuous high-quality base length less than 75% of the length of the tags. The clean tags then underwent chimera detection using the UCHIME algorithm until all chimeras were removed and effective tags were obtained (*Edgar et al., 2011*). All of the sequences were clustered into operational taxonomic units (OTUs) of " ≥ 97%" similarity using the UPARSE version 7.0.1001 pipeline (*Edgar, 2013*) and the representative sequences were classified into organisms by applying a naive Bayesian model using the RDP classifier version 2.2 (*Wang et al., 2007*) based on the SILVA database (version v132 ) (*Pruesse et al., 2007*).

## Statistical analysis

The taxonomic composition for each cluster was evaluated at the phylum and genus level. Data related to bacterial community were statistically analyzed using SPSS version 17.0 (SPSS, Inc Chicago, IL, USA). The significance of herbage nutritional composition and bacterial taxa was determined using the independent-samples T test. The bacterial community function was predicted using Tax4Fun software (version 0.3.1) (*Asshauer et al., 2015*). The significant KEGG metabolic pathways were screened using STAMP software (version 2.1.3), the statistical method was two groups Welch's $t$-test, the type was two-sided and the confidence interval method was Welch's inverted (0.95). The OTU rarefaction curves were calculated and plotted in QIIME. All alpha diversity was calculated in QIIME and graphed by Origin version 8.0 (OriginLab®, Northampton, MA, USA). The principal coordinates analysis (PCoA) with unweighted UniFrac distance at OTU level was plotted in R version 3.5.0 and metric distance were calculated by Wilcoxon rank sum test. The unweighted pair-group method with arithmetic means (UPGMA) classification tree was plotted by mothur version 1.41.1 (*Schloss et al., 2009*). Results are reported as means ± standard error (SE). The effects were considered significant at $P < 0.05$.

## Network analysis and identification of putative keystone taxa in TWAs and NPDDs

To understand the interaction of gut microbiota between TWA and NPDD, phylogenetic molecular ecological networks (pMENs) (http://ieg4.rccc.ou.edu/mena) were constructed based on Random Matrix Theory (RMT)-based methods. The protocols of network construction were described previously (*Li et al., 2016*). Briefly, only the OTUs that were presented in more than half of all samples were selected to in each TWAs and NPDDs gut microbiota genus. In order to compare the topological characteristics of bacteria network between TWAs and NPDDs, the pMENs were calculated with the same threshold (0.76). For each network, in the above website, the window "global network properties" was used to calculate total nodes, total links, positive links, negative links, average degree (avgK), average clustering coefficient (avgCC), centralization of betweenness (CB) and density (D).

**Table 2  The dry matter digestion between TWAs and NPDDs.** TWAs refers to Tibetan wild asses, NPDDs refers to natural pasture domestic donkeys.

| Indexes | Groups | | P value |
|---|---|---|---|
| | TWAs | NPDDs | |
| AIA in faces | $0.41 \pm 0.05^b$ | $0.47 \pm 0.01^a$ | 0.000 |
| AIA in herbage | $0.17 \pm 0.04^b$ | $0.24 \pm 0.02^a$ | 0.274 |
| Dry matter digestibility | $58.90 \pm 8.21^a$ | $50.46 \pm 3.71^b$ | 0.043 |

**Notes.**
AIA, Acid-insoluble ash.

It was worth noting that a positive correlation might relate to mutualism, commensalism or parasitism, while a negative correlation might result to competition, predation, etc. (*Faust & Raes, 2012*). The window "module separation and modularity calculation" was then used to calculate the value of within-module connectivity (Zi) and among modularity connectivity (Pi). Finally, the visualization of these community co-occurrence networks was made with Cytoscape 3.3.0 (*Shannon et al., 2003*).

A module typically contains many nodes, which are tightly connected to each other in one group, while having only few connections outside the group. The values of Zi and Pi are indicators of the connectivity of each node, and thus are often used to determine the topological role of these nodes (*Deng et al., 2012*). According to the valises of Zi and Pi, these nodes can be classified into four categories (*Zhou et al., 2011*), including peripherals (In the modules, the OTUs have few outside connections, Pi < 0.62 and Zi < 2.5), connectors (OTUs that connect modules, Pi > 0.62), network hubs (OTUs that highly connected within entire network, Pi > 0.62 and Zi > 2.5) and module hubs (OTUs that highly connected within modules, Zi > 2.5). Connectors, network hubs and module hubs were considered as the putative keystone taxa in a microbial community (*Li et al., 2017*). The keystone of gut microbiota in TWAs and NPDDs was identified based on the above values.

## RESULTS

### Analysis of food nutrition composition and dry matter digestibility between TWAs and NPDDs

As shown in Table 1, the mainly plant-based diet for TWAs were *Kobresia tibetica* and *Kobresia pygmaea*, while for NPDDs were *Kobresia humilis* and *Kobresia capillifolia*. There was no significant difference in nutrient composition except ash content ($P < 0.05$). In addition, the dry matter digestibility was measured by acid-insoluble ash method. As shown in Table 2, AIA in the faces of NPDDs were significantly higher than TWAs group ($P < 0.05$), whereas, dry matter digestibility in NPDDs were significantly lower than TWAs group ($P < 0.05$).

### Sequencing and classification

Illumina sequencing yielded 2,540,672 raw reads of 16S gene sequences. After quality filtering, a total of 2,395,867 effective tags were obtained (Table S1). As shown in Fig. S1, 12,923 operational taxonomic units (OTUs) at a cutoff of 97% similarity were generated.

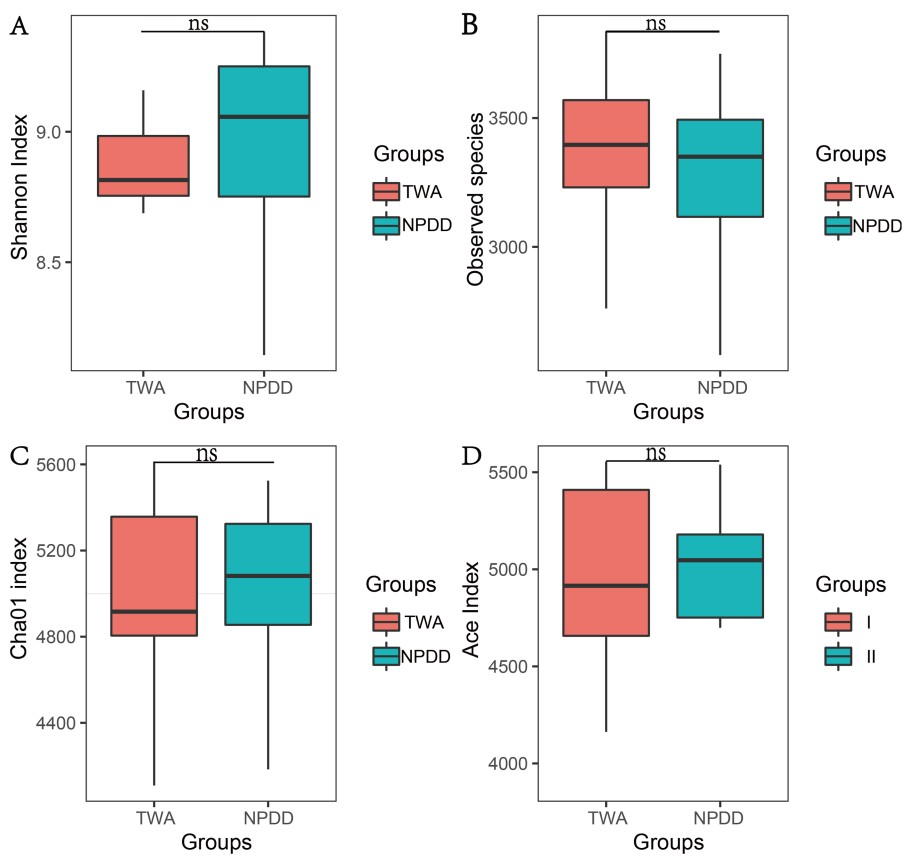

**Figure 1  The statistical significance of alpha diversity estimators between TWAs and NPDDs.** TWA, Tibetan wild asses; NPDD, natural pasture domestic donkeys. NS means no significant difference. (A) Shannon index; (B) Observed species; (C) Cha01 index; (D) Ace index.

Among these OTUs, the number of unique OTUs in TWAs and NPDDs was 2948 and 2816, respectively, and 7159 OTUs were shared by all samples, accounting for 55.40% of the total OTUs.

## Diversity analysis of gut microbiota in Tibetan wild ass and domestic donkey

To determine alpha diversity, we calculated the Chao1, ACE, observed species estimates and the Shannon diversity index. As shown in Fig. 1 and Fig. S2, no significant differences were found in these two groups for the Shannon index, Chao1, ACE estimates, the number of observed species and Faith's phylogenetic diversity (PD). The OTU level rarefaction curves of diversity estimators reached a plateau at the minimum sequence number of 50,000 tags (Fig. S3). With regard to beta diversity, PCoA analysis showed that the two groups of bacterial communities differed significantly (Fig. S4A) ($P = 3.02\text{E-}05$). To assess the overall similarity, UPGAMA with classification tree analysis was conducted as shown in Fig. S4B. It is obvious that the bacterial community of TWAs was completely different from that of NPDDs.
## Significance analysis of gut bacterial community abundance between TWAs and NPDDs

At the phylum level illustrated in Fig. 2A, *Bacteroidetes, Firmicutes, Verrucomicrobia,* and *Fibrobacteres* constituted the four dominant phyla in all samples. The average ratio of *Firmicutes/Bacteroidetes* was 0.80 and 0.74, respectively. The significance analysis of the top 10 phyla of microbial communities between TWAs and NPDDs is shown in Fig. 2B and Table S2. The relative abundance of *Bacteroidetes, Firmicutes, Cyanobacteria,* and *Synergistetes* in TWDs was significantly higher than in NPDDs, whereas *Lentisphaerae* in TWAs was significantly lower ($P < 0.05$). No significant differences were detected for the relative abundances of *Verrucomicrobia, Fibrobacteres, Spirochaetaes, Proteobacteria,* and *Planctomycetes* between the two groups ($P > 0.05$).

At the genus level, a total of 15 genera with proportions above 1% were detected, as shown in Fig. 3A. The *Rickenllaceae_RC9_gut group, Fibrobacter,* and *Treponema_2* accounted for the largest proportions in all samples. Differentiation analysis of the 15 genera is presented in Fig. 3B and Table S3. The proportion of *Ruminococcaceae_NK4A214, Phascolarctobacterium, Coprostanoligenes_group, Lachnospiraceae_XPB1014_group,* and *Akkermansia* in TWAs was significantly higher than in NPDDs, whereas the proportion of the *Lachnospiraceae_AC2044_group* in NPDDs was significantly higher than in TWAs ($P < 0.05$). No remarkable difference was detected in the relative abundance of the *Rickenellaceae_RC9_gut_group, Fibrobacter, Treponema, Ruminococcaceae_UCG-010, Christensenellaceae_R-7_group, Prevotellaceae_UCG-001, Anawrovorax, Prevotellaceae_UCG-003,* and *Prevotellaceae_UCG-004* between the two groups ($P > 0.05$).

To study the main differences in gut microbiota between groups TWA and NPDD, linear discriminant analysis (LDA) combined effect size measurements (LEfSe) were used to identify the discriminative features of bacteria in the TWAs and NPDDs (Fig. S5). The taxonomic units of *Campylobacter, Campylobacteraceae, Campylobacterales, Epsilonproteobacteria,* and *Campylobacter_hyointestinalis_subsp_hyointestinalis,* which belong to the phylum *Proteobacteria,* were significantly enriched and revealed different biomarkers in TWAs, whereas no specific taxonomic units were enriched in NPDD.

## Predicted metabolic pathways and functions of the bacterial microbiota between TWAs and NPDDs

A total of six types of biological functional pathways in KEGG level 1 were detected (Fig. 4A). The metabolism pathways had the highest relative abundance, with more than 60% of the total reads in each group. At KEGG level 2, a total of 37 metabolic pathways were detected in the gut samples (Table S4). As shown in Fig. 4B, 18 gene families were remarkably different and the relative abundance of amino acid metabolism, energy metabolism, glycan biosynthesis, and the biosynthesis of other secondary metabolites in TWAs were significantly higher than in NPDDs ($P < 0.05$). As for membrane transport and cell motility, NPDDs had higher expression abundances than TWAs ($P < 0.05$). At KEGG level 3, the principal component analysis and partial least squares discriminant

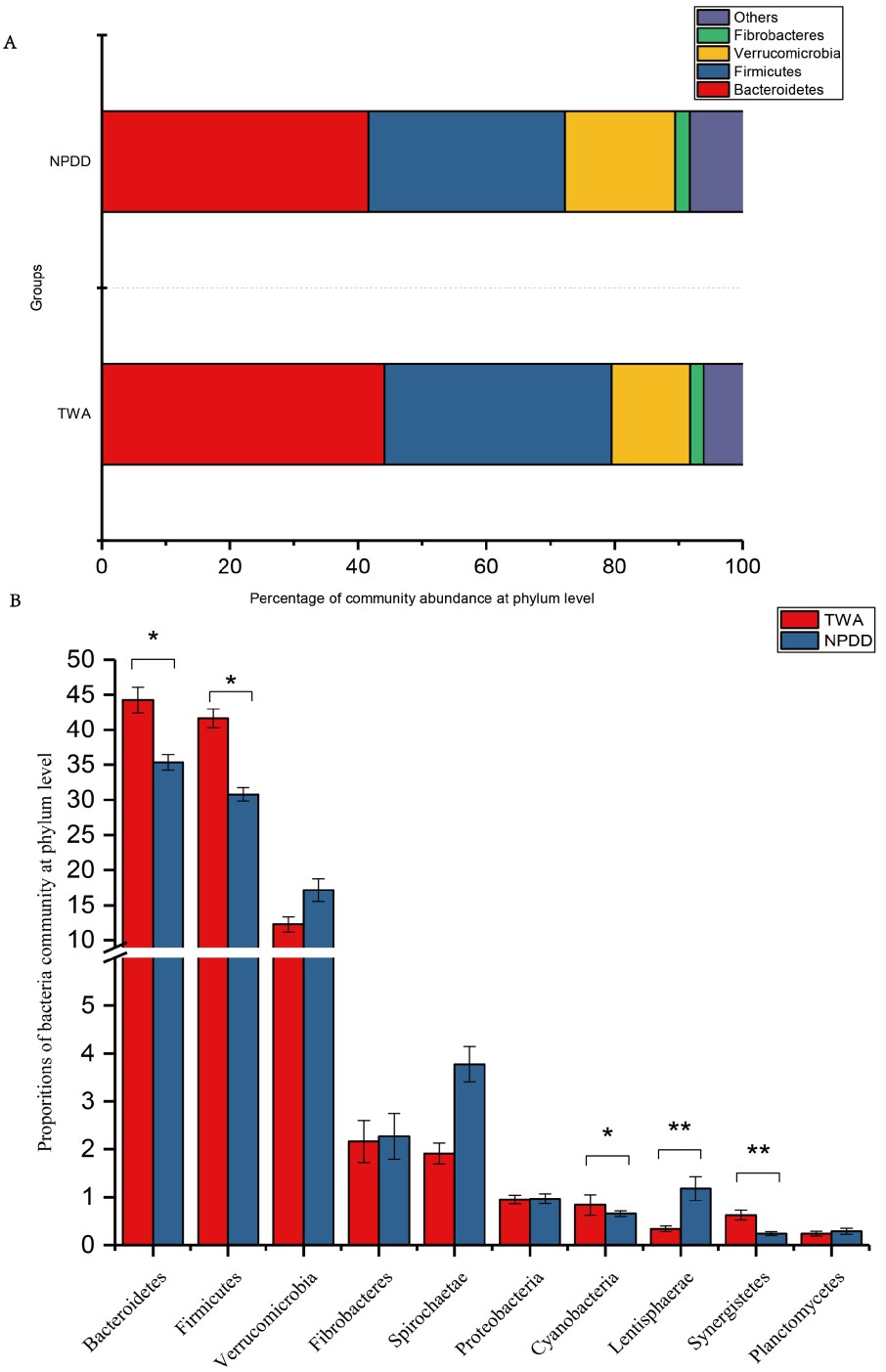

**Figure 2 Bacterial community relative abundance at the phylum level of each group.** (A) Bacterial community abundance bar plot at the phylum level. (B) Significance of the top 10 bacterial community abundance percentages at the phylum level. Each phylum that shares annotations was significantly different ($P < 0.05$). The error bar meant the value of standard error. TWA, Tibetan wild asses; NPDD, natural pasture domestic donkeys.

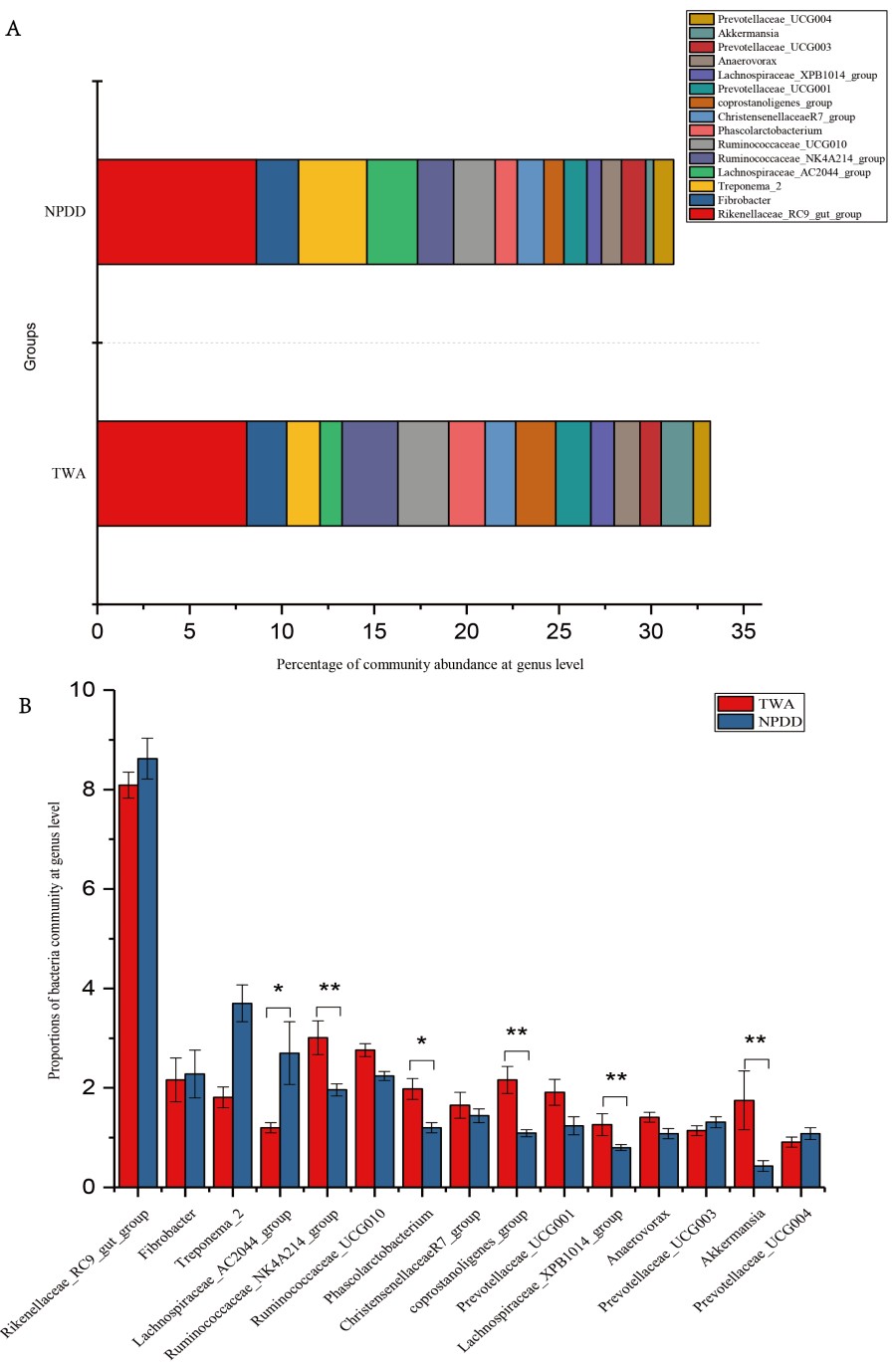

**Figure 3** **Bacterial community abundance at the genus level.** (A) Bacterial community bar plot at the genus level. (B) Statistical significance of the top 15 bacterial community abundance percentages at the genus level. The error bar meant the value of standard error. TWA, Tibetan wild asses; NPDD, natural pasture domestic donkeys.

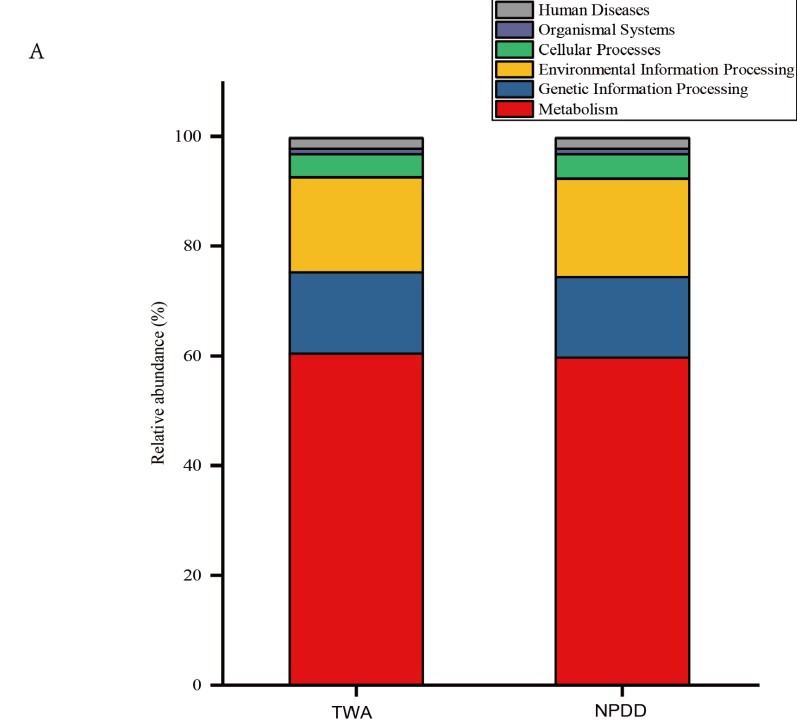

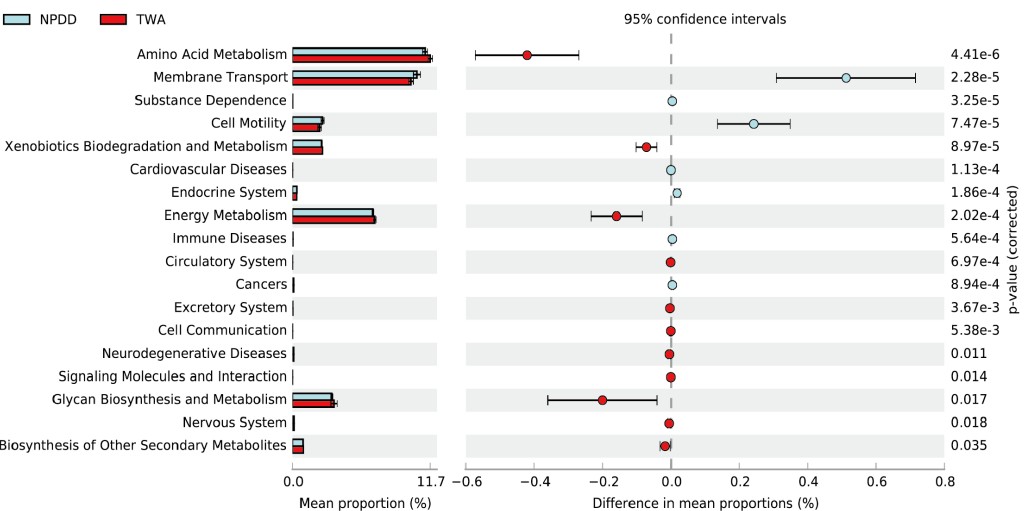

**Figure 4 Metabolic pathway prediction at various KEGG levels between TWAs and NPDDs.** (A) The six types of biological functional pathways in KEGG level 1. (B) The 17 significant metabolic pathways in KEGG level 2. The "*p*-value (corrected)" is the false discovery rate (FDR) value. TWA, Tibetan wild asses; NPDD, natural pasture domestic donkeys.

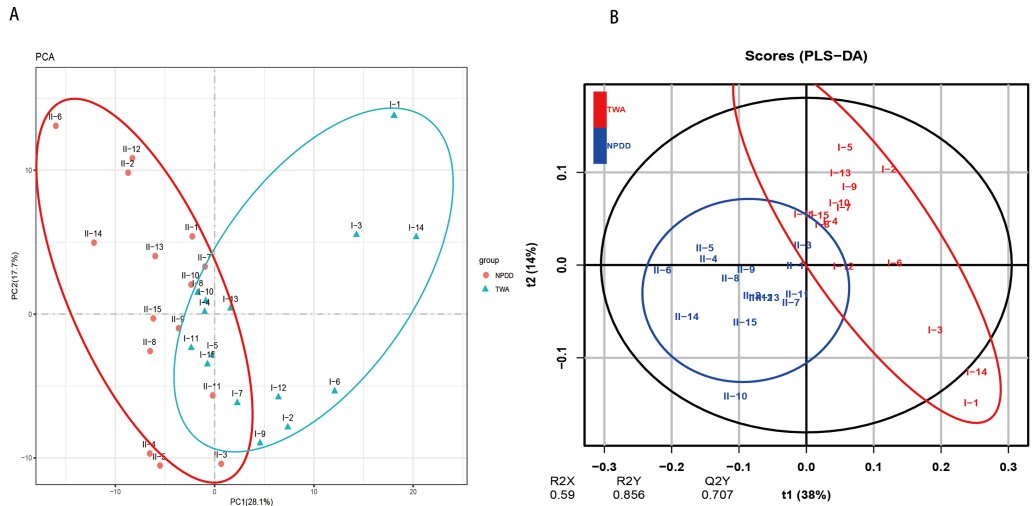

**Figure 5** **The principal component analysis (PCA) (A) and partial least squares discriminant analysis (PLS-DA) (B) of metabolite pathways between TWAs and NPDDs.** TWA, Tibetan wild asses; NPDD, natural pasture domestic donkeys.

analysis revealed clear differences between the two groups (Fig. 5). As shown in Table S5, 40 metabolite pathways were significantly different ($VIP > 1$ and $P < 0.05$), of which ten metabolite pathways belonging to amino acid metabolism were enriched in TWAs ($P < 0.05$). A total of ten metabolic pathways belonged to carbohydrate metabolism of which seven metabolites were enriched in TWAs, whereas the remaining three metabolites (starch and sucrose metabolism, amino sugar and nucleotide sugar metabolism, and lipopolysaccharide biosynthesis) were enriched in NPDDs ($P < 0.05$). In addition, five metabolic pathways belonged to the metabolism of cofactors and vitamins, of which two metabolites were higher in TWAs. For the remaining 15 metabolic pathways, 11 had a higher relative abundance in TWAs.

## Network analysis of bacterial communities and putative keystone taxa

As shown in Fig. 6, when using the same threshold (0.76), a total of 207 and 104 links were identified for TWA and NPDD. Notably, for TWAs, the percentage of positive correlations (149 out of 207) was dominant, while negative correlations (55 out of 104) were the predominant in NPDDs. Interestingly, we found the values of total links, avgK and D were higher in TWAs than those in NPDDs, indicating a relatively simpler network in NPDDs. The lower avgCC in NPDD indicated that the bacterial network was mainly loose node groups. Moreover, TWA had a higher level of CB than NPDD, implying that TWA harbored a higher frequency of centralized.

The putative keystone genus in bacterial networks was identified based on the Pi and Zi values (Table 3). Three key module hubs and two connectors were identified in NPDD. Among these OTUs, module hubs were affiliated with *Firmicutes*, *Cyanobacteria* and unclassified bacteria, and the connectors were *Anaerovibrio* and unclassified bacteria. In contrast, four connectors and no module hubs were identified in TWAs. Among these

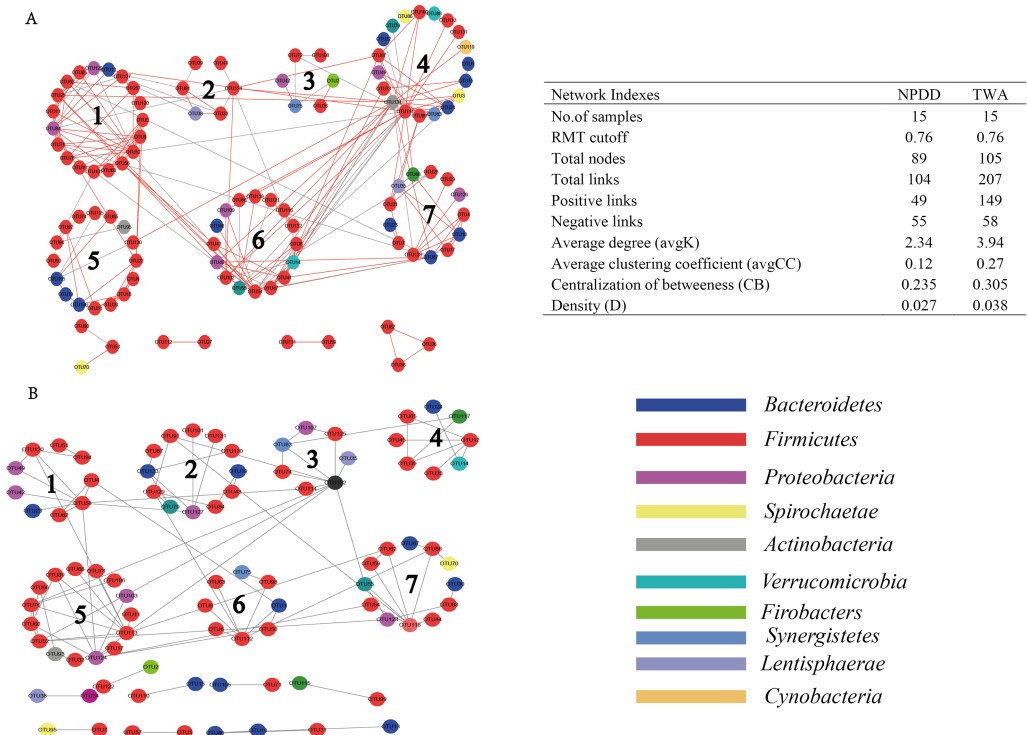

| Network Indexes | NPDD | TWA |
|---|---|---|
| No.of samples | 15 | 15 |
| RMT cutoff | 0.76 | 0.76 |
| Total nodes | 89 | 105 |
| Total links | 104 | 207 |
| Positive links | 49 | 149 |
| Negative links | 55 | 58 |
| Average degree (avgK) | 2.34 | 3.94 |
| Average clustering coefficient (avgCC) | 0.12 | 0.27 |
| Centralization of betweeness (CB) | 0.235 | 0.305 |
| Density (D) | 0.027 | 0.038 |

*Bacteroidetes*
*Firmicutes*
*Proteobacteria*
*Spirochaetae*
*Actinobacteria*
*Verrucomicrobia*
*Firobacters*
*Synergistetes*
*Lentisphaerae*
*Cynobacteria*

**Figure 6   Comparision of gut bacterial networks between TWAs and NPDDs.** (A) The gut bacteria of TWA. (B) The gut bacteria of NPDD. Highly positive correlations are indicated by red color and negative correlations by gray color. TWA, Tibetan wild asses; NPDD, natural pasture domestic donkeys.

OTUs, two connectors were affiliated with *Firmicutes* and *Actinobacteria* and the others were the genus *Oscillospira* and *Arthrobacter*. Notably, these keystone genera in network showed relatively low abundances in gut bacterial communities.

## DISCUSSION

TWAs and NPDDs are herbivores species on the QTP, however, scant research has been conducted on their gut microbiota diversity and composition. To the best of our knowledge, this is the first study to compare the gut microbiota composition and function of TWAs and NPDDs. We found that the Shannon-Wiener index and ACE and Chao1 estimates, as well as the number of observed species between TWAs and NPDDs were not significantly different, suggesting that altitude and grassland types did not affect the alpha diversity of gut microbes. A previous study had shown that diet was one of the factors that affected gut microorganism diversity (*Daniel et al., 2014*). In our study, there was no significant difference in herbage nutrient composition between TWAs and NPDDs in different meadow types (swamp meadow and alpine meadow, respectively), which was consistent with the results of the alpha diversity analysis. In terms of biological classification, TWAs and NPDDs are animals of the same genus but different species; however, there was no significant difference in the alpha diversity of gut microbes, suggesting that species differences did not affect the

**Table 3** **The taxonomic identification of putative keystone genus in network of TWAs and NPDDs bacterial community.** *Fir Firmicutes, Act Actinobacteria, Cya, Cyanobacteria.* TWAs refers to Tibetan wild asses, NPDDs refers to natural pasture domestic donkeys.

| Species | OTUID | Average relative Abundance[a] | Taxnomy | Abundance ranking[b] | Node type | Pi | Zi |
|---|---|---|---|---|---|---|---|
| TWAs | OTU67 | 0.0662 | *Unclassified* | 53rd | Connector | 0.6914 | 0.4115 |
| | OTU114 | 0.0046 | *Unclassified* | 116th | Connector | 0.6950 | 2.2368 |
| | OTU124 | 0.0034 | *Oscillospira* (*Fir*) | 121th | Connector | 0.7347 | 0.0000 |
| | OTU134 | 0.0027 | *Arthrobacter* (*Act*) | 128th | Connector | 0.7160 | 1.7847 |
| NPDDs | OTU58 | 0.0043 | *Cellulosilyticum* (*Fir*) | 118th | Module hubs | 0.5185 | 2.5628 |
| | OTU112 | 0.0035 | *Erysipelatoclostridium* (*Fir*) | 126th | Module hubs | 0.4490 | 2.5019 |
| | OTU118 | 0.0023 | *Oscillatoria* (*Cya*) | 137th | Module hubs | 0.2449 | 2.9542 |
| | OTU54 | 0.0517 | *Anaerovibrio* (*Fir*) | 68th | Connector | 0.6667 | 0.8452 |
| | OTU114 | 0.0032 | *Unclassified* | 130th | Connector | 0.6400 | 2.1243 |

**Notes.**
[a]The average relative abundance of the OTUs in TWAs and NPDDs.
[b]The abundance ranking of the OTU in TWAs and NPDDs in TWAs and NPDDs according to order from high to low

alpha diversity of bacteria community, similar result was also found in the previous study on wild and domestic yak (*Liu et al., 2019*). As to beta diversity, PCoA results indicated that the composition of gut microbiota was significantly different in these two groups. For a better understanding of the similarity of their gut microbiota, UPGMA methods were applied and the diagram revealed that the fecal microorganism communities in TWAs occupied their own phylogenetic tree branch which clearly distinguished from the fecal microorganisms in NPDDs. Therefore, we concluded that species, altitude, and meadow type can affect the intestinal bacterial structure; however, the determination of which of these played the leading role in shaping intestinal flora requires further studies.

To assess the composition of the bacteria community, we first analyzed at the phylum level. In our study, *Bacteroidetes* was the most abundant phylum followed by *Firmicutes* in both TWAs and NPDDs, constituting more than 80% of the total bacteria content. This has been confirmed by former studies on intestinal microbial diversity in mammals (*Mariat et al., 2009*; *Middelbos et al., 2010*; *Sahu & Kamra, 2002*), as these microorganisms facilitate the digestion of cellulose and hemicellulose in the diet (*Wu et al., 2016*). The relative abundance of *Bacteroidetes* and *Firmicutes* in TWAs was significantly higher than in NPDDs, indicating that the cellulose and hemicellulose decomposition capacity was stronger in TWAs than in NPDDs under similar forage nutritional composition, and the higher dry matter digestion capacity in the large intestine in TWAs also supports this claim. Moreover, the ratio of *Bacteroidetes* and *Firmicutes* affected energy acquisition and body fat accumulation in humans and mice (*Turnbaugh et al., 2006*). Previous studies have shown that a decrease in *Bacteroidetes* was strongly linked to an increase of fat in host tissue in mice and humans (*Ley et al., 2006*; *Turnbaugh et al., 2006*). However, many recent studies have found controversial results. *Schwiertz et al. (2010)* found that *Firmicutes*/*Bacteroidetes* ratio changed in favor of the *Bacteroidetes* in overweight and obese adults. With regard to perissodactyla animals, however, scant research has focused on the relationship between gut microbiota and obesity in TWAs and NPDDs. By observing the

body size of TWAs and NPDDs, we speculated that the lean body size of TWAs might be related to the high *Bacteroidetes* content in their intestinal microbiota. At the genus level, 15 core genera whose relative abundances were more than 1% were detected in all gut samples; these genera constituted 32.43% of the total genera reads and were mainly affiliated with the phylum *Firmicutes* (nine genera) and phylum *Bacteroidetes* (four genera). The others were affiliated with the phyla *Spirochaetes* (one genus) and *Verrucomicrobia* (one genus). These results were also reported in ruminant research on animals such as Yaks and Tibetan sheep (*Li et al., 2019*; *Xue et al., 2017*), indicating that although TWAs and NPDDs belonged to the genus *Equus* and were different from ruminants, their microorganism flora communities were actually quite similar. In addition, an average of 54.43% genera presented as unclassified, which suggested that this group of organisms may be a part of the core bacterial population, warranting further investigation in the field on *Equus*.

Recent studies have shown that changes in gut microbiota composition are associated with host disease (*Costa et al., 2012*; *Qin et al., 2012*; *Sekirov et al., 2010*). *Bacteroidetes,* a basal microbe, is reported to be the most abundant phylum in healthy people (*Eckburg et al., 2005*) and captive TWAs (*Gao et al., 2019*); a decrease in its relative abundance is associated with chronic diarrhea in humans (*Eckburg et al., 2005*) and could be a predictor of an animal's health. In our study, it accounted for a large proportion in TWAs (44.12% of sequences), indicating that TWAs may have a superior capacity to stave off intestinal diseases. At the genus level, *Akkermansia* accounts for the highest proportion in *Verrucomicrobia*. Researchers have found that *Akkermansia* currently contains a single species, namely *Akkermansia muciniphila,* which is an important probiotic that contributes healthy mucus-related microbiota composition (*Belzer & De Vos, 2012*; *Liu et al., 2014*). In addition, *Akkermansia muciniphila* can be used as a probiotic to prevent obesity and type 2 diabetes (*Everard et al., 2013*; *Qin et al., 2012*). In our research, TWAs had a higher distribution proportion of *Akkermansia muciniphila*, which may be important to the maintenance of intestinal health; however, further study is necessary to verify this assertion.

The metabolites of gut microbiota play a pivotal role in maintaining physiologic and metabolic homeostasis in their hosts (*Human Microbiome Project C, 2012*). In the present study, the metagenomic function prediction showed that the two predominant intestinal microbiota in TWAs and NPDDs were related to ABC transporters and the two-component system (Table S6). Previous research (*Xiong et al., 2014*; *Zeng et al., 2017*) has confirmed that ABC transporters constitute one of the largest known protein families and are widespread in bacteria, archaea, and eukaryotes, and the two-component system is a signal transduction system that senses developmental and environmental stimuli (*Podgornaia & Laub, 2013*). Therefore, it is reasonable that these KOs were found to be in high abundance in the intestinal microbiota of TWAs and NPDDs. Moreover, statistical comparisons showed that 40 different metabolic pathways exhibited significant differences (Table S5). Among them, 29 pathways had richer concentrations in TWAs than NPDDs, mainly amino acid metabolism, carbohydrate metabolism, and energy metabolism. A large number of various microorganisms inhabit the intestinal tracts of monogastric animals that are symbiotic with the hosts. Gut bacteria produce a range of metabolites that provide nitrogen sources for themselves and amino acids for the hosts through the fermentation

of protein and nitrogen compounds. In our study, the high concentration of amino acid metabolic pathways in TWAs implied that in the context of high altitude and forage shortage, the gut microorganisms and hosts coevolved and exploited high bioavailability to efficiently utilize nitrogen compounds in forage, ultimately with microorganisms providing themselves a nitrogen source and in return producing various amino acids to maintain normal life activities for the hosts. Research has revealed that carbohydrates in forage can be fermented by intestinal microorganisms and the metabolites include certain short chain fatty acids (SCFAs) which can stimulate the intestinal mucosa, improve intestinal immunity, and promote energy absorption in the hosts (*El Kaoutari et al., 2013*; *Jha & Leterme, 2012*). In our study, the concentration of carbohydrate metabolic pathways in TWAs had an advantage over NPDDs, which meant that microorganisms in the gut of TWAs could more efficiently promote cellulose and hemicellulose fermentation. As for energy metabolism, we found that the citrate cycle in TWAs was significantly more active than in NPDDs, indicating that the intestinal bacteria of TWAs could provide more energy for their own metabolic activities and were therefore beneficial for microorganism fermentation of forage in the intestine.

There are complex interactions in gut microbiota, such as mutualism, competition, prediction and commensalism (*Faust & Raes, 2012*). Understanding the interactions and their responses to environmental is an important goal in ecology. However, defining a network structure in microbial communities is especially challenging due to their vast diversity and as-yet uncultivated status. Thus, we used the network analysis to investigate the genus interactions according to their mutual competition (negative correlation) and mutual benefit (positive correlation). In our network analysis, the TWAs were predominated by higher positive correlations, while NPDDs were predominated by negative correlations. We speculated that plant-based diet in the gut of TWAs was more conductive to fermentation through the more corporation interactions than that of NPDD. In addition, we found that in TWAs, the total links, avgD and D were relatively higher than that of NPDDs, indicating that TWAs harbored a more complex ecological network, mostly also more metabolite. More complex interactions could build a more stable metabolic network environment for the microbial population. Further study is needed to confirm this hypothesis.

A lower avgCC revealed that bacterial communities are largely composed of isolated nodes or loosely connected node groups, reflecting lower functional redundancy (*Sun et al., 2013*). In our study, the avgCC in TWAs was relatively higher than NPDDs, indicating that the latter may harbor lower function redundancy than the former. The previous study had revealed that centralized nodes represent keystone species, which may influence the "information" flow among microbes (*Bissett et al., 2013*). In addition, these nodes play an important role in connecting many other microbes and are generally considered as important control points in the networks a low level of CB, as a result of a low frequency of centralized nodes (*Bissett et al., 2013*). Thus, a higher CB in TWAs, indicated that the microorganism had different status in networks, implying that the loss of one or several keystone species may affect the stability of the whole bacterial community. Our result also showed that NPDDs harbored more keystone genus than TWA. Here, a total of three module hubs and six connectors were observed in TWAs (four connectors) and NPDDs

(three module hubs and two connectors, respectively). However, none of them was the top 10 most abundant genus. The most abundant module microorganism OTU67 was at the 53rd position, which indicated that the keystone genus in network not necessary the dominant microbes in the gut microbial community. However, the topological features are related to sequencing depth and different data processing. In our study, we only remained the OTUs that were present more than half of all samples, and this data processing may influence the topological features in bacteria network.

## CONCLUSIONS

This study described the dry matter digestion, compositions, diversity and function of bacterial communities in Tibetan wild asses and domestic donkeys. Based on 16s rRNA sequencing data, we found that the gut microbiota of TWAs was superior to that of NPDDs in bacterial community composition, function, and potentially high resistance to disease risk under similar forage nutrition intake. In addition, high dry matter digestion ensured wild assess use the forage efficiently in high altitude on the Qinghai-Tibet Plateau. As the only wild species of perissodactyls on the Qinghai-Tibet Plateau, the protection of Tibetan wild asses is of great significance to maintain the balance of the ecosystem and the harmony between human and animals. From the perspective of molecular biology, we studied the difference of gut microbiota between Tibetan wild asses and domestic donkeys. These results may help us understand the assembly of bacteria community in the gut of wild and domestic animals and provided a theoretical basis for the adaptability of the intestinal flora of Tibetan wild asses at high altitudes.

**Abbreviations**

| | |
|---|---|
| **QTP** | Qinghai-Tibet Plateau |
| **TWAs** | Tibetan wild asses |
| **NPDDs** | natural pasture domestic donkeys |
| **4N HCL** | 4 molar hydrochloric acid |
| **AIA** | acid-insoluble ash |
| **Avgk** | average degree |
| **AvgCC** | average clustering coefficient |
| **CB** | centralization of betweenness |
| **D** | density |

## ACKNOWLEDGEMENTS

We express our sincere gratitude to Dr Qi Li for his warm-hearted help of the research group during the sample collection.

## Funding

This work was supported by the National Key Research and Development Program of China (No. 2018YFD0502301), the National Natural Science Foundation of China (No. 31402120), the Key R&D and transformation plan of Qinghai Province (No. 2019-SF-153), and "The Dawn of West China" 2018 Talent Training Program of CAS by Dongdong Chen. The funders had no role in study design, data collection and analysis, decision to publish, or preparation of the manuscript.

## Grant Disclosures

The following grant information was disclosed by the authors:
National Key Research and Development Program of China: 2018YFD0502301.
National Natural Science Foundation of China: 31402120.
Key R&D and transformation plan of Qinghai Province: 2019-SF-153.
2018 Talent Training Program of CAS.

## Competing Interests

The authors declare there are no competing interests.

## Author Contributions

- Hongjin Liu conceived and designed the experiments, performed the experiments, prepared figures and/or tables, authored or reviewed drafts of the paper, and approved the final draft.
- Xinquan Zhao, Liang Zhao, Linyong Hu, Tianwei Xu, Na Zhao, Xiaoling Zhang, Dongdong Chen, Fuquan He and Xin Chen performed the experiments, authored or reviewed drafts of the paper, and approved the final draft.
- Xueping Han analyzed the data, authored or reviewed drafts of the paper, and approved the final draft.
- Shixiao Xu conceived and designed the experiments, performed the experiments, authored or reviewed drafts of the paper, and approved the final draft.

## Animal Ethics

The following information was supplied relating to ethical approvals (i.e., approving body and any reference numbers):

In this study, all animal care procedures were consistent with the guidelines from the Institution of Animal Care and Ethics Committee of the Northwest Institute of Plateau Biology, Chinese Academy of Sciences (NWIPB20160302).

## Field Study Permissions

The following information was supplied relating to field study approvals (i.e., approving body and any reference numbers):

This project is carried out with the permission of the local government (Qinghai Province, China). Zhang Chaoyuan, the Deputy Director of the China Qinghai Provincial

Science & Technology Department, and Zhao Xinquan the Academic Dean of Northwest Institue of Plateau Biology Chinese Academy of Sciences gave verbal authorization to allow us to carry out the experiment, so official documents and field permits are not required in this study.

## Data Availability

The sequencing data for the 16S rRNA genes are available in the NCBI Bioproject Archive: PRJNA553267. The raw data are available in the Supplemental Files.

## Supplemental Information

Supplemental information for this article can be found online at http://dx.doi.org/10.7717/peerj.9032#supplemental-information.

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
