# Peer review of "Comparative study of gut microbiota in Tibetan wild asses (Equus kiang) and domestic donkeys (Equus asinus) on the Qinghai-Tibet plateau"

_PeerJ, doi:10.7717/peerj.9032_

## Round 0.1 · original submission · Minor Revisions

Dear authors,

Thank you for submitting your work to PeerJ.

Please, read the comments from the two reviewers bellow and adjust the MS accordingly to clarify the issues raised, or provide a strong rebuttals, and submit a revised version of your MS.

Specifically, confirm that chloroplastic signals have been removed from the dataset, making it clear on the MS, and I agree with both reviewers that some associations between the data and phenotypes should be less speculative.

Kind regards,
Thiago Parente

Reviewer 1 ·

Basic reporting

The quality of this manuscript is overall sound. The language is clear and well-organized, despite many trivial English which can be fixed. The Introduction and Discussion sections cited multiple literatures which are relevant to the topic to my point of view.

Experimental design

The research is original. The question is clearly defined. As authors indicated, this is the first study of TWA gut microbiome and it is helpful with the protection of this species. The Methods section contains sufficient details for readers to understand the entire protocol, though with some caveats as detailed below. The procedures are fair.

Validity of the findings

The authors shared the raw sequencing data (Fasta format). The results were sufficiently described. The conclusion was well stated based on their observations. However I have two major concerns of the validity of the results, which would require further clarification, as stated below.

Additional comments

This manuscript reports a study of the gut microbiome of Tibetan wild asses in comparison to that of domestic donkeys. The authors analyzed 16S rRNA V3-V4 sequencing data collected from fecal samples. They compared the taxonomic and functional profiles inferred from the DNA data. They found differential abundances and compositions. Based on the observations they proposed hypotheses to explain the differential physiological and ecological features of the two species, especially considering the ability of dry matter digestion.
The quality of this manuscript is overall sound. However I have the following concerns.
Two concerns are major and require the authors to carefully clarify: 1) Fig. 2B: What metric does the error bars represent (in this figure and other figures)? They look very small. However according to Fig. S3B, I doubt if they are this small, if they represent standard errors. It would also be great if the authors provide actual p-values instead of asterisks. The abundances are not very different, but the differences are significant according to the authors. The only explanation is that variance is low (as the error bars suggest).
2) Table 3: OTU118 Bromus tectorum is a plant, not a cyanobacterium. Is this a typo? Also see Line 219: “The relative abundance of Bacteroidetes, Firmicutes, Cyanobacteria, and Synergistetes ...” Either way, I want to ask the authors to double-check and clarify whether chloroplastic signals have been removed from the dataset. If not, this is a big problem because donkeys are herbivores and their fecal samples must have high plant contamination.
The following concerns are relatively minor but the authors still need to clarify seriously.
The first use of “TWA” and that of “4N HCL” in the abstract (line 21) are not explained.
The statement that “wild asses were superior to that of domestic donkeys in dry matter digestion, gut microbial community composition and function” (line 38) is not objective. One can’t assert that one microbiome is “superior” to another.
Line 143: Please state QIIME version and the main QIIME script / command used for filtering.
Line 149: Please state which SILVA release was used.
Line 155: Please state Tax2Fun version.
Line 156: Please state STAMP version.
Line 161: Please spell the full name of SE (standard error) at first usage.
Figure 1: alpha diversity: the authors should also try Faith’s PD, which is phylogenetic, since they used the phylogenetic metric UniFrac for beta diversity.
It appears to me that the distribution of alpha diversity is bimodal in both groups. What are the outliers at the low end? Why their microbiomes are so un diverse?
OTU filtering. Why the number of species can reach a few thousands? Can the authors explain?
Fig. 4B: What is “p-value (corrected)”? I guess it is FDR correction, but the authors need to clarify. In particular, which FDR-correction method was used and what was the population size (n)?
Table 1: “P valve” => “P value”
Table 3: “Unclassfied” => “Unclassified”
Line 210: “With regard to beta diversity, PCoA analysis showed that the two groups of bacterial communities differed significantly (Fig. S3A) (P < 0.05).” How was this p-value calculated? Sorry if I missed anything, but I did not see relevant description in the Methods section.
In the LEfSe analysis (line 238), all taxonomic units appear to be the same bug: Campylobacter. This is fine and I just want to note it.
Line 394: “lose” => “loss”
Line 314: “… we speculated that the lean body size of TWAs might be related to the high Bacteroidetes content in their intestinal microbiota.” This is a bold speculation, however it is not necessarily inconsistent with PeerJ’s peer-review guideline. Therefore I won’t complain but just note it to express my reservation.

·

Basic reporting

Overall, the article is well-written. However, some aspects can be improved.
i) Abbreviations:
- Line 18: Add the TWAs abbreviation after “Tibetan Wild Asses”, since the abbreviation is used afterward in line 21.
- Line 220: Abbreviation used was “TWD”. Shouldn’t it be TWA?

ii) Some rephrasing
- Line 28: “the two dominant phyla of Bacteroidetes and Firmicutes”. The preposition “of” should not be used here. The whole sentence could be rewritten.
- Line 38, 79, 408 : The use of “were superior” is not suitable in the cases in which it was used. Sentences with “were superior” should be rewritten.
- Line 279: It is not clear why the word “Although” was used in this sentence.
- Lines 335 - 336: “Researchers have found that Akkermansia contains a single species”. Since more Akkermansia species may be identified in the future, it would be better if the sentence was: “Akkermansia currently contains a single known species”.

iii) Grammatical errors
- Line 25: “were” should be “was”.
- Line 31: add “in” after “higher than”.
- Line 275: “Firmicutes” was misspelled.

iv) Literature references
- Lines 309-315: Although Turnbaugh et al. 2006 found correlations in the decrease of Bacteroidetes and obesity, many recent studies have found controversial results. These controversies could be briefly addressed. The following review state many details about this theme: Castaner et al. The Gut Microbiome Profile in Obesity: A Systematic Review. Int J Endocrinol. 2018

Experimental design

- Line 119: The authors should add a reference or explanation for the use of the equation to measure Dry Matter Digestibility.
- Lines 165-177: positive/negative correlations should be explained here since it will be explored throughout the article.
- Lines 264-265: could have a brief reminder of what positive/negative correlation means.

Validity of the findings

- Lines 210-211: The PcoA analysis was based on what taxonomic rank (e.g. genus, family, phylum)? Considering the barplot based on phylum in Fig S3B, TWAs and NPDDs gut microbiota seem very similar.
- Figures 2B and 3B: It is curious that Spirochaetae and Treponema did not show significant differences between TWA and NPDD. I advise the authors to double-check this.
- In Supplemental files, the authors could add tables with mean abundances of TWAs and NPDDs taxonomic profiles, as was done for the metabolic pathways.
- Line 248: Some of the 18 gene families presented in Fig 4B don’t appear to be “remarkably different” between TWA and NPDD. In fact, some of these gene families were identified with less than 0,1% abundance (based on Table S2). Therefore, it would be good to use a minimum abundance threshold and measure the significance afterward.
- Line 332-334: The association of NPDDs having more abundance of Spirochaetes and being more likely to suffer from an intestinal disease is too speculative and should be removed.

---

## Round 0.2 · accepted · Accept

I am happy to bring you this good news during this hard time we are all facing.

Please, during the proof reading of the edited version of your MS, consider the typos indicated by Reviewer 2.

Stay safe.

Reviewer 1 ·

Basic reporting

no comment

Experimental design

no comment

Validity of the findings

no comment

Additional comments

I appreciate that the authors have made dedicated point-to-point responses to my comments as well as the other reviewer’s. Both of my previous major concerns have been well-addressed. I have no more concerns and hereby recommend for publication.

·

Basic reporting

see below

Experimental design

see below

Validity of the findings

see below

Additional comments

I thank the authors for taking my comments and suggestions into account and I think the paper is now suitable for publication.

But I still found a few typos:

Line 235: "sinificant" -> "significant"
Line 267: "VIP" -> "P"
Line 328: "reagard" -> "regard"
Line 410: "53th" -> "53rd"
Line 418: "commubities" -> "communities"